COMMENT

# Developing a new generation of scientist communicators through effective public outreach

Sean M. Mackay [1], Eng Wui Tan [1] & David S. Warren [1✉]

Science disengagement amongst school children remains a global challenge, leading to calls for more scientists to engage with the public. Here the authors discuss how a voluntary, flexible program can enhance graduate attributes in addition to addressing barriers to public engagement.

## Introduction

**The need for public outreach**. Do you remember what first sparked your interest in science? Was it a school experience, a particular teacher, a lesson, a color, or even a smell? Perhaps, like the New Zealand Nobel laureate Alan MacDiarmid, it was simply a book, The Boy Chemist, read as a youngster[1]. Whatever the event, it kindled a passion and led to a life-long interest and a career. Therefore, it is very worrying to see that around the Organization for Economic Co-operation and Development (OECD), children are becoming increasingly disengaged from science, with many losing their interest and confidence at an early age[2]. Indeed Anderhad et al.[3] suggest that an interest in science may never actually be constituted during a pupil's primary school years. One answer to this disengagement would appear to lie in the evolving and diverse field of science communication. But what is science communication? In addition to the wide plethora of electronic and social media-based content, publicly funded bodies such as museums and universities (Beacons for public engagement; NCCPE. https://www.publicengagement.ac.uk/) commonly run programs offering an enormous range of activities to increase engagement with science and make it cool to be a nerd[4]. This diversity gives rise to a fundamental issue for someone setting up a new program; where to find a model that can be used as a starting point?

In the current political climate, the combination of public discussion of scientific controversies and the opinion that researchers in publicly funded systems should be obligated to participate in public engagement has created a perceived need/requirement for scientists to engage with the general public in a relatable way[5–9]. Yet meaningful engagement with the general public is often hampered by the public perception of a scientist and who they are; they are seen as separate from the wider community, creating a barrier. This is illustrated in a fascinating study from Yale (Cultural Cognition project-home. http://www.culturalcognition.net/)[10], which has demonstrated that people are capable of holding two sets of beliefs about controversial issues: an expert opinion that they understand and accept, and one that they adopt with their peer group or

[1] Department of Chemistry, University of Otago, Union Place West, Dunedin 9016, New Zealand. ✉email: dwarren@chemistry.otago.ac.nz

community. Scientists need to be perceived as part of the general public as opposed to being inside the Ivory Tower of academia[11]. The importance of the removal of this disconnect is highlighted by the ASPIRES project in the UK, where work by Archer et al.[12–15] shows that aspirations of students are shaped not by their achievement in science but, rather, how they position themselves under the influence of a wide range of factors, such as ethnicity, social class, and family, pointing to public perceptions of scientists and who becomes a scientist.

Our opinion is that at the simplest level, irrespective of what is done, science community engagement (outreach) should be considered as something done to influence and improve the attitude and awareness of the wider community towards scientists and science in general.

**Barriers to faculty participation**. Debates on social media and the literature highlight many issues surrounding public engagement, discussing the role scientists should play, and even questioning what public engagement means[7,16,17]. In 2005 Andrews et al.[18] reported on Participation, Motivations, and Impediments to scientists taking part in public outreach in the USA. They found the three top barriers to staff and students were time, lack of information about outreach and lack of value/lack of support within the institution. The value put on research outputs essentially meant that outreach had a low priority. This makes outreach difficult to justify in spite of the acknowledged benefits to the staff and students who took part in the activities. These finding were echoed in a later study of biologists and physicists[19], which reported that 74% of respondents listed one or more significant barriers to their ability to do science outreach. These barriers were attributed to one of three broad elements: scientists and their lack of skills in this area; the academy and the focus on research output, with its related time constraints and lack of training in non-research related activities; and a combination of public knowledge and interest around science. In the UK a study by the Royal Society about science communication[20] reported similar barriers: time taken away from research being the major barrier, followed by; disapproval by other scientists; disruption of a career; lack of funding for outreach. Funding seemed especially important with 81% of the respondents saying that they would be encouraged to do more outreach if they could bring more money into their department. Burchell[21] also reported that lack of confidence and/or training is a significant factor in public engagement, as well as professional stigma and lack of reward, with animplied lack of value, from institutions, especially around promotion criteria.

In general, whenever community engagement is mentioned within our institutions, it is almost guaranteed that new academics are discouraged from participating by managers worried that the time demands are a barrier to career progression[21]. Those who take up the gauntlet are generally young staff who deliberately move towards science communication as a career[22], or more senior and established academics who can afford the time commitment required.

**Brief description of our program and its approach**. Here, we discuss our approach to delivering an engaging schools outreach program and try to show how some of these barriers can be overcome and the benefits to all parties involved. We believe that by engaging with schools and communities, scientists not only help stimulate young children's interest in science, but we also equip our graduate students with a wider range of skills than those gained from the usual experience in tertiary study, developing a generation of scientists who have the skills for effective public engagement. At the heart of our approach to working with

schools and communities is the New Zealand Māori concept of ako, meaning a reciprocal or two way learning process. We recognize that as scientists we can learn as much from the community as they can from us, moving away from the more traditional deficit model of outreach often seen as a justification for public engagement. This is widely recognized as a more powerful way to work with communities around the world than the traditional model of scientists providing a one way transmission of knowledge[21,23–25]. All of our outreach is free of charge to schools given our target demographics, this approach also ensures that there is joint learning and both parties benefit from these exchanges of knowledge and culture.

For us, successful outreach means developing a positive attitude towards chemistry in young school children. In many countries, including New Zealand, school pupils start to disengage with science as early as 8–12 years old[26–29], especially at schools in rural and less affluent areas where science is not perceived as part of everyday life or is not seen as a relevant career[13,14]. By working with these communities, we can start to break down traditional barriers between scientists and the general public. We feel this is an important step in the process of rectifying science disengagement, and working in schools means we are the visitors and the school is the host, generating a more relaxed atmosphere. This is an important consideration given a study of the impact of our program on a small rural school has shown that a visit by university scientists can be stressful for pupils in our target demographic[30].

Our philosophy of outreach is that it should build on children's innate curiosity of their world using a wide range of hands on activities, during long-term partnerships with schools. We approach this through a program involving a coordinator who mentors a team of chemistry student volunteers. Typically we use 2 or 3 students per visit, with up to 15 active students over an average year, to develop and deliver 80–100 outreach events throughout the year. It is valued by our partners that this coordinator is both a trained teacher and a scientist. The approach itself is nothing new, many outreach programs around the world use students to facilitate such activities and have ex-teachers acting as coordinators. However, we have made a conscious effort to develop long term relationships with a small number of target schools, with two of our current partnerships are in their 11th year. We make multiple visits, to schools over a year, in some cases we may visit a school up to ten times in the year. This approach has two main benefits. Firstly we develop very close relationships with classes, teachers and families, to the point that we are invited to attend school functions such as the end of year prize giving as guests. Secondly, it allows our own students flexibility, by providing a diversity of schools and activities. Students can postpone their involvement in outreach for significant periods of time and still work with the same schools when they return to the program. Running activities at different times each week, as well as evenings[30] and weekends also allows undergraduate volunteers to participate, permitting long-term membership of the outreach team; some students have spent up to 7 years with the program and delivered up to 2000 h of outreach. Students develop skills, confidence, and the ability to run their own section of the program with minimal support from the coordinator. A further benefit of such a flexible system in a range of schools is that the coordinator can mentor the students and suggest new challenges as skills develop (Fig. 1)[31]. We feel this program maximizes the impact for our students while minimizing possible negative aspects of an outreach program such as time management, frustration and the feeling of lack of support[32].

Although it may not be obvious at first glance, the most important part of our three segment strategy for a session (see

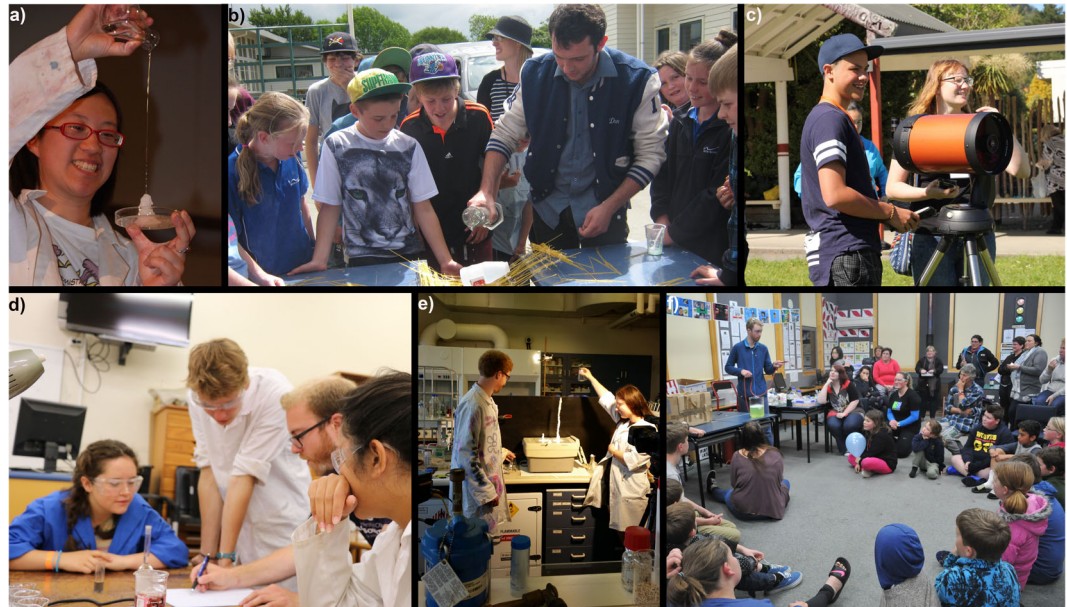

**Fig. 1 Images to illustrate the progress of students through our outreach program. a** Demonstrating activities, showing participants how to grow a stalagmite; **b** front of house delivery running a "spaghetti bridge" competition; **c** planning outlines of activities, working on ocean acidification with Māori students. **d** Developing new ideas for teaching high school students, here delivering activities around nanoparticle research. **e** Developing new ideas from scratch, here leading a team to film teaching resource videos. **f** Designing, planning, and running a program, here an evening program called Science for Supper. The images should be taken to represent typical steps in the "career" of our outreach student rather than being seen as activities all students take part in.

Methods section) is the "show and tell" at the beginning. During this segment, we are constantly asking questions of the group while rarely giving straight answers. We are trying to find out the depth of knowledge and understanding that the group has about the topic. This is especially important in new schools with new teachers, and we frequently modify our approach or expected outcomes to the hands-on activity based on the response to this first segment. The approach reflects the influence of the teaching background of the coordinator, and also the influence of people such as Rosalind Driver in the UK, who has argued that teachers, i.e., outreachers, need to understand how pupils interpret a situation and understand what they bring to a classroom in order to get the best outcomes[33]. This approach is described by Eleanor Duckworth and co-workers as "Critical Exploration"[34–37]. We strongly recommend that scientists who wish to meaningfully engage with children through outreach or community engagement read Duckworth's book The Having of Wonderful Ideas[34].

## Methods

A typical year of our program starts in January with two "hands on" residential camps on our Dunedin campus. These are university initiatives that offer projects involving 12 h of engagement with a group of senior high school pupils. One of the camps (Hands on at Otago (https://www.otago.ac.nz/hands-on-at-otago/index.html) has a general University experience ethos, while the other (Otago University Advanced School Science program (University of Otago Advanced Science Academy, (O. U. A. S. S. A.) https://www.otago.ac.nz/ouassa/index.html)) is intended to support pupils from rural and low income communities during their final year, and offers a year-long program for both the students and their schools. Both camps typically involve activities developed by senior outreach students based on current research interests from within the department. This has led to the creation of engaging education programs centered on cutting-edge research in a diverse range of fields such as nanoscience, crystallography, hydrogel materials[31], and the herbicidal

properties of NZ native plant extracts[38]. These projects give school pupils a sense of the research process, how science links to applications, and the scope of research that takes place within the local University. These camps offer an ideal first step for department students who are interested in outreach, and we frequently use students conducting summer research projects on campus, or those living locally over our main summer holidays. In the past, we have had ex-outreach students who have gone on to work in science communication return to work with our team and try to pass on ideas and advice to new team members. Within the department several research groups have contributed towards these projects, recognizing and valuing the creation of education resources from their research and developing, overcoming their own lack of expertize in this process.

Our regular school program, aimed at pupils under 12 years of age, starts in late February after schools open for the new academic year. Our preference is to work within schools to become "part of the community", and we mainly work with rural and low income schools for reasons outlined in the introduction. Typically we allow 90 min per group but use shorter sessions for younger children. Each session has three segments. We start with a demo/discussion related to the upcoming hands on activity (e.g., a Q & A about dissolving things and growing sodium acetate stalagmites-critical exploration), followed by a hands-on activity for the students (e.g., making saturated solutions of alum or copper sulfate, filtering off undissolved material and setting up containers to grow crystals), and finally, a summary Q & A session to wrap up and consolidate the activity with more critical exploration.

One of the benefits of extended engagements is that we can follow up the progress of their crystals as they grow and deliver an activity with a related concept during a future visit. A fundamental requirement of the activities is that they are flexible and can be adapted to a variety of situations and age groups (Fig. 2). This is important as it may not be feasible to take several different activities to one school when we are traveling long distances to

| | | | |
|---|---|---|---|
| **Location** | Residential camp OU campus | Workshop in Taiwan conference centre | Rural school Malaysia/NZ |
| | 2 days (final year high school) | 3 hours (final year high school) | 90 minutes (10-14 years old) |
| **Synthesis of a tough-gel** | ✓ | | |
| **Measuring Young's modulus** | ✓ | ✓ | |
| **Modelling with tough slime** | ✓ | ✓ | ✓ |

**Fig. 2 An example of how outreach topics can be adapted to both location and "audience."** In this case, the concept of nano-clay tough-gels can range from synthesis of gels or measuring Young's modulus of pre-prepared samples, to exploring concepts using PVA/borax slime modified with cornstarch or metal oxide powders. This exploring allows an element of play as the participants get to vary the mass of additive and feel how the gel changes as the mass is varied. Although seen as "play", it is an important step in developing an appreciation of how the particles interact and modify the gel properties.

**Table 1 Summary of the outreach carried out in 2019.**

| Description | Total number of schools | Total number of visits | Total number of students | OU students (average per activity) | OU staff (average per activity) | Venue |
|---|---|---|---|---|---|---|
| Primary/intermediate | 7 | 37 | 2120 | 3 3 | 1 | In school |
| Secondary | 8 | 8 | 260 | 1 | 1 | On campus |
| "One offs" | 8 | 8 | 90 | 2 | 1 | On campus |
| International | 16 | 8 | 1500 | 7 | 3 | In school |
| Residential camps | 40 | 3 | 70 | 5 | 2 | On campus |
| Other staff | 9 | 12 | 300 | 0 | 1 | Off campus |

rural schools and spending a whole day and work with several classes. Therefore, a single activity needs to be amenable to a range of ages, abilities, and requirements. Some of these activities have been used with international partners and prove to be equally effective in engaging school pupils across language and culture barriers[39–41].

Activities are usually designed by the outreach coordinator and/or a senior outreach student, who then leads the delivery supported by the rest of the team. A great example of this approach is something we call Science for Supper[42]. Originally a collaborative idea between the Chemistry and Physics outreach coordinators inspired by comments from parents, it was developed into its final form by two senior students, who designed the activities and developed the pedagogy. It runs in the early evening and has become central to our main program to involve the wider community in an informal situation. It has proven to be a great chance to talk with parents and often generates conversations about their own science experiences at school, as well as being. It is one of the most popular ways for undergraduate students to begin their outreach journey, the use of evenings allows them more opportunity to be involved.

Given the time constraints within the secondary sector our high school outreach mainly centers around providing support for schools carrying out a Year 13 extended investigation (an option assessment in the final year of high school in NZ that involves 12 h of lab work). For this we provide access to labs, technical support, material and apparatus for up to 50 students per group over a two day period during semester breaks. The

range of activities runs from titrations, e.g., Vitamin C or chloride ions, to using atomic absorption spectroscopy for metals analysis or colorimetric determination of Vitamin B6 levels in fruit. We also support a local citizen science project (Healthy Harbor Watchers) that monitors the water in the Dunedin harbor for a range of health indicators. This latter is a great example of how Universities can support local projects with minimum input. The project is run through local schools by local teachers, using our lab space at weekends to carry out the analysis of water samples from the harbor. Our role is to provide the facilities, general glassware and some materials, and one of our outreach students will generally participate during visits.

Since 2015 we have also run an International outreach program during our mid-winter break in late June and early July, working with the Madame Curie High School Chemistry Camp in Taiwan, and across Malaysia with a range of teacher training institutions. The background to its establishment is complex, but the drive for us was to provide a challenging program for involving our most experienced leaders. These leaders have an opportunity to build their own teams of promising junior members, creating a core of students with wide experience and expertize to take over the leadership roles in future years. All stages of planning, including selection of the rest of the team, involved the leaders with the outreach coordinator. It has become one of the most sought after outreach activities for students involved in the outreach program and has provided some outstanding learning opportunities for both our students and staff[40,41].

**Table 2 A rubric developed to benchmark student development through outreach.**

| Level | Statement | Descriptor | Example |
|---|---|---|---|
| Supporter | Aware of outreach | The student occasionally does outreach when encouraged to. Does not connect their experience and their role as a scientist | Attends outreach as part of a group, or when specifically asked to join in |
| Novice | Responds to outreach | The student actively attending outreach, pro-active & asking for new opportunities but expresses their involvement in terms of their own benefits. | Attends regularly as a volunteer, attends without being asked, starts to talk of enjoyment, works with small groups but not confident to talk to whole class or lead an activity |
| Learner | Values outreach and shares experiences | The student sees the impact of outreach on the community group and gives their role a value beyond the learner/expert dialog and sees the role they are playing in the development of positive attitudes in their partners/learners | Recognizes the response in the community to outreach, talks about excitement in kids, starts to feel confident in front of a group, starts to lead activities or whole class |
| Facilitator | Balances their role in outreach with their role as a scientist | starts to advocate for outreach as a desired activity for their peer group, sees the impact on their learning | Starts to recruit/advocate for outreach, leads sessions, has developed an internal dialog that they are comfortable with |
| Leader/ initiator | Outreach becomes a recognized and valued part of their role as a scientist | The student looks to initiate projects independently, Community engagement and its advocacy becomes part of their expectations for their future career in science | Starts to organize whole events, develop their own program, start to see the development of others as important and reflects on their role as mentor |

## Results

Like any other University outreach program, our program varies from year to year. The raw numbers for 2019 can be seen in Table 1. It should be recognized that compared with other countries, or even other areas of New Zealand, the population density of the southern part of the South Island of New Zealand is low.

The total number of students is calculated by heads per visit. Thus, if we see one student three or four times over the year they count as three or four students. Staff can also include alumni who frequently work with us for the summer residential camps and the International programs, and in New Zealand the primary/ intermediate sector covers 5 to 12 years old. The *one off* category covers a range of visits to the University by, for example, Māori and Pacific Island pupils as part of programs aimed at raising educational aspirations within groups that are under-represented at University in NZ. *Other staff* are outreach activities carried out by other members of academic staff that the outreach program supported with materials/suggestions and or training (something that has grown more popular with time).

Although outreach is a voluntary program it is important that students have opportunities to grow and develop through their experiences. This, which means the outreach coordinator and outreach leaders need to be able to benchmark the progress of younger team members. Recently we have begun to develop a rubric that can be used in a formative manner to establish the degree of development students are showing. In general terms, the descriptors can be seen in Table 2, starting with students who come along occasionally and help out, leading through stages of increasing awareness of the role that they are playing within the team and in the development of positive attitude by the school pupils, through to the leaders who initiate their own programs but more importantly recognize the development of more junior members of the team and act as mentors. This is still in development but it gives an idea of how students can develop during their time on the program. It was developed from literature around service learning courses in the USA[23,43] and the much smaller number of similar descriptions around outreach programs[7,31,44–46].

In a focus group interview in 2014, as part of a research project into the impact of outreach within the Division of Sciences at Otago University, a group of leaders ($n = 5$) discussed their impressions around their involment in outreach with a research assistant. The following, unpublished, data represents anonymised transcripts from this 60 min discussion. Many of the comments illustrate their development through the levels of the rubric in Table 2.

There were several comments around the role they played in gaining interest of the school pupils

"…you see their eyes light up and you think, actually you have got through to them and have had an impact on their learning and for me personally that is huge…"

"…but I think is sparking an interest in science from such an early age and seeing them understand. And when they start asking questions back I think that is really neat because you know that they are getting it and they are interested because they want to know more."

"…our job in outreach is […] to get people interested in science but if you can use that and you can help someone develop into a better person overall then I would say that is being a good mentor, going the extra step…"

"…one of the fundamental things about science in general is that it's an exercise in curiosity so if we can spark people's curiosity and creativity then we don't have to have taught them what the boiling point of water is. That to me is the mark of a good lesson, or one of them."

They talk about developing a rapport with the groups

"…one of the most important things, mentoring things that we can do is to sort of allow them to explore their ideas, to create an atmosphere where they are prepared to be wrong but as long as they are giving ideas, it's ok."

"I would say that is probably a skill that outreach has taught us as well, getting a sense of what a student's background is and how they are going to respond to us…"

The importance of debriefing and self-reflection on their own practice;

"…we all travel together in a van and that's quite a good time to sort of debrief I guess about what worked and what didn't. How could we improve this? And I think that's really valuable."

Two of them described the development of a program in a local school;

"…input from everybody is the way it got built up. We had the lessons and then someone would say, I think it would work better this way so we would do it that way."

Another member then added

"I can't say I specifically added to any lesson plan in a formal way but there were changes I made on the fly that weren't documented."

Indicating their increasing confidence and comfort with developing their own style and approach.

The theme of teamwork and mutual effort was a common one, for example;

"…and I think actually that's how it goes. Someone will do their practical part of the demonstration or the experiment and then someone else will do the talking and that's sort of how we work together I guess."

"If you have never done it then it can be very demanding and you don't know how to handle it, it can be quite, not scary, not something that you don't want to be involved with, but you see all these other people do it so you think it can't be that bad so you need to have the courage to give it a try yourself and put the explanations with it as well."

"Someone could go from cleaning glassware one week to presenting a whole lesson the next week if they wanted to because the support is always there. There are always the people who have done it before who can help them out and yeah, so no one is kind of locked into a certain role…"

Finally recognizing the role of the leaders is more than just the work in the classroom

"To be honest the most important thing about the leadership role with schools though is not so much the teaching when you are already there but the actual organizing to get ready to go to the lesson. Getting all the chemicals ready, making sure you know who your team is, what their roles are, organizing transport to and from the University and School…"

but also an awareness of the feeling within themselves of the progress they made

"…and saying an explanation that you had scripted and it doesn't work, the kids don't get it so you have to come up with something on your feet and after a while of doing that you realize you do the right amount of preparation but you can also back yourself to be up there on stage. You know you know the chemistry, you know you know the content and then it's all about the performance, it's all about connecting with the audience. I think that felt like quite a big thing. We could just—if you think back to what we were like at the start, if someone had asked us to do that we would have been like, nope, not a chance."

Anticipating the development of the rubric described above during a discussion about achievements of others in the program

"That would be handy if you were trying to put a team together and you could look over a list and say, ok I have these people who have presented, these people who are very good support people and I need one of each…"

One of the more junior members of the group talked about the impact it had in building their confidence when another interrupted to talk about a job interview they had just had (at this stage two of the students had just left the chemistry department)

"…I guess the gut feeling is that outreach helped me with just being able to deal with any question they throw at you and because.. I mean kids throw out all sorts of questions and you have just got to deal with them. And I think that's something that I have been able to take with outreach…"

this elicited the following comment about self-efficacy from another

"…the confidence you gain from really anything always seeps into every part of your life. Your work life, your social life, just everything."

They then discussed the flexible, informal approach that we have within the program to student learning and the role that a script plays in supporting presenters but then also showing a deep insight into how others will develop past this

"…so you have got to make sure the formal structure is like a safety net, everyone can start with that and then some will move past it and some will stick to it."

As the session ended they also talked about what more the University could do for outreach programs which was summed up by this statement

"I guess an awareness of the contribution we are making and we feel that we are making anyway to schools and to the general public and some of the impact it has for the universities, the perception as a whole."

Finally, in 2019 one of this group, now an alumnus who continues to work with the program on occasion, was interviewed by a science communication intern working with the program to develop our social media platform gave the following quote;

"What outreach gave me, through engagement with the community, combined with the skills gathered in my undergrad has meant I have never had to apply for a job but rather have had them offered to me. The outreach combined with my undergrad put me in that position."

## Discussion

Recently, a report from the UK suggested that repeated, long-term engagement may be part of the solution to student disengagement: pupils who receive outreach for 2 to 3 years are more likely to take science A-levels; a correlation which suggests a more positive attitude towards science[47]. Our own experiences agree with this approach: monitoring by teachers has shown that our long-term programs appear to create a more positive attitude towards science in 12-year-old students compared with national statistics, although data from the ASPIRES project in the UK does note that this positive attitude falls off during the later school years. A research project by a local school principal shows that long-term relationships with communities build increased trust in the scientists and increased awareness of science within the wider community[30]. However, such engagements require resources and, more importantly, regular time commitments which many staff within a research-focused environment cannot

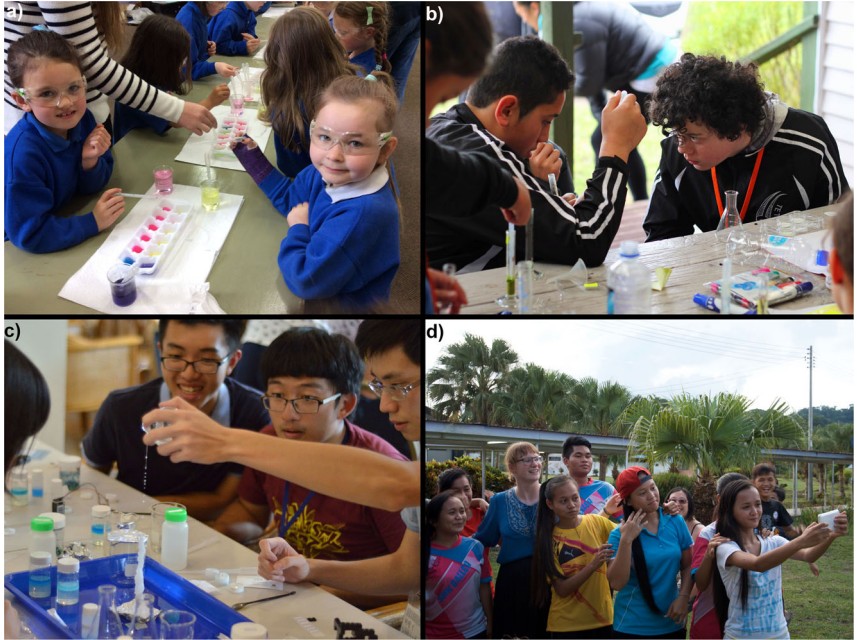

**Fig. 3 Photos from a year in our program. a** Primary school pupils in rural NZ explore colors using plants from the school garden and dilute acid and alkali. **b** Making plant extracts to test as natural herbicides during activities at a marae (māori meeting house). **c** Students in Taiwan making a sodium acetate stalagmite during a crystal workshop. **d** End of a visit and farewells at a remote rural school in Malaysia.

afford[18,21,39–42]. We believe that the system described here, with a centralized coordinator and student volunteers, overcomes some of these issues. It also allows other staff access to a program that they can take part in, or use as a resource for their own visits, cutting down on planning time and providing training. Many of our resources were sourced from teaching materials used by one of us (DSW) during a teaching career or are readily available on-line, for example the Salters Chemistry club (http://resources. schoolscience.co.uk/Salters/) or The Royal Society of Chemistry "Classic Demonstrations" book[48]. Volunteers, both staff and students, are encouraged to develop these ideas for the school they work with and also keep within their comfort zone, expanding as they grow in confidence and looking for their own sources of resources and ideas. As a result, the department culture has become one of support and encouragement for the program.

Finally, what are the benefits of outreach to the organization itself that would encourage support? Aside from good public relations and an increasingly frequent requirement for public accountability placed on research funding, there are numerous advantages gained by students and institutions from a long-term outreach program[7,23,49]. The Association of American Colleges and Universities defines graduate success in the modern world as more than retention, recruitment and graduation rates; activities such as outreach are described as "high impact educational practices"[50]. Like many others, our own University has a strategic plan with a series of core values, a number of which can be met through a voluntary outreach program, without adding to workload. Indeed, community engagement is increasingly seen as core business for Universities. Through effective public outreach, students develop graduate attributes[51] such as self-confidence, global awareness, and communication skills, which can be difficult to develop in a purely science-based curriculum within a research-centered institution. Participation in an outreach program enables students to work as a team, trusting and depending on each other, developing skills that produce well-rounded graduates

in a way that just is not possible in a lecture or laboratory environment, and this spills over into everyday life in a department[49,52]. Furthermore, students make some of the best ambassadors for science and their institution, they represent a more diverse, younger, accessible, approachable and outgoing group than public expectations of scientists; qualities which have added tremendous value to programs we have run over the past 10 years working with children across New Zealand, Taiwan, and Malaysia (Fig. 3)[39–42].

In our experience, a large part of the excitement and engagement that occurs during outreach activities is not solely due to the fact that the children are "doing" science, but rather comes from the positive attitudes of the students facilitating the activities. In contrast, senior, more authoritative, figures such as an academic staff members, can often create a more formal attitude among school pupils creating a barrier to their engagement.

### Future directions
As community engagement increasingly becomes part of a scientists required output[7,53] the fact that many scientists are still learning these new skills while finishing academic training, or alongside a research career, is a major challenge. We can address this as a community by developing a new generation of graduates that are equipped to work with the wider community. Our experience over the past 10 years, working with thousands of children across three countries, has led to the belief that well designed, flexible outreach programs that engage communities from the bottom-up create substantial value to both the chemists and audiences who participate. We aim to inspire children towards a life-long interest in science, while enabling the next generation to develop the skills required to become effective ambassadors for chemistry and overcome the perceived barriers that exist between the scientific community and the wider public[30]. This is a compelling case for increased investment in outreach programs in research-centered institutions, and our

program has attracted several international visitors to spend time working with us. As a result, we are now looking at developing small micro-credentials for students who wish to get a more formal recognition of their progress through outreach. It is envisaged that this will involve the development of a teaching program across the Division of Sciences, aimed at the many students who are involved in outreach in a wide number of projects, at the last count there were upwards of 170 outreach programs within the Division. As well as having a short practicum, these courses will focus on self-reflection, "critical exploration" as a teaching tool and the use of qualitative data to understand outcomes.

## Data availability

We are happy to enter into discussion with interested parties about our experiences and findings over the 11 years that this program has operated; please contact the corresponding author.

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

## Acknowledgements

We wish to acknowledge the support of the following: University of Otago chemistry department for ongoing funding and support for the program; The University of Otago Division of Sciences for financial support through their Strategic Initiative fund; our International partners, the Madam Curie Chemistry Camp Foundation in Taiwan, IPG Batu Lintang and IPG Tun Abdul Razak teacher training colleges in Kuching, Malaysia and Sultan Idris Education University, Malaysia for financial and material support; The Royal Society of Chemistry and the American Chemical Society for funding towards our International program. We acknowledge the role played by all our partner schools both in NZ and overseas without whom none of this could have happened, and the dedication and hard work by the many students who have participated in the program over the last 10 years. Finally, SMM gratefully acknowledges a University of Otago Doctoral Scholarship and financial support from the New Zealand Ministry of Business, Innovation and Employment.

## Author contributions

The original concept for the paper was a joint one between all the authors and all three authors have been involved with the editing process. The manuscript was drafted by S.M.M. and D.S.W. and then edited by all three authors. D.S.W. is the coordinator of the outreach program described in this manuscript, S.M.M. has worked extensively on the program both as a student and an alumnus in New Zealand and abroad. E.T. was the supervisor of S.M.M. for his Ph.D and has first hand experience of the impact of the program on students in the department.

## Competing interests

The authors declare no competing interests.
