## [Peer Review File · Communications Chemistry]

Reviewers' comments:

Reviewer #1 (Remarks to the Author):

This is an interesting paper, but it suffers from a very serious lack of organization. In its present form it is merely a report of an interesting program, but if properly organized it could be helpful to other universities that might be interested in organizing their outreach efforts in a similar manner. I think that the article needs a major revision, and that it should be organized to include the following sections:

1. Introduction

1.1. The need for public outreach

1.2. Barriers to faculty participation

1.3. Emerging trends in science communication

1.4. Brief description of the unique aspects of this program

2. Methods

Detailed description of a typical year of this program, including personnel responsibilities, costs, organizational diagrams, etc.

3. Results

Outcomes from a year of this program, including number of school visits, number of children served, and number of students, faculty, and staff involved in this effort.

4. Discussion

5. Future directions

Reviewer #2 (Remarks to the Author):

This article describes an outreach programme in New Zealand and advocates for scientists, in particular chemists, to do more science outreach. While the outreach programme described is very valuable, I am not sure how it adds to the body of evidence in the field. The programme is quite typical of much outreach work in the UK, and I would expect some empirical evaluation research evidence to back up these assertions rather than just feedback and opinion. The paper would fit well as a practice paper in some science communication journals, but I do not think it fits with a Nature journal.

The literature referenced comes from letters to Science/Nature or Twitter, with very few empirical literature. The paper would benefit from researching the science communication literature, or collaborating with a science communication researcher. P1 The facts stated are not quite true - interest in science has been shown to remain high - it is aspirations to pursue a career in science which tail off with age, or identification with science as a hobby or with relatable role models. Please refer to and draw on the work of the Aspires Study in the UK, and Science Capital as a concept, with work by Louise Archer. Please also refer to some science communication literature for a definition, linking this to public engagement e.g. Rowe and Frewer 2005. You may also wish to draw on literature from LENSscience in Auckland, or Meet the Scientists in Southampton, UK.

To whom it may concern,

Developing a new generation of scientist communicators through effective public outreach.

Sean M. Mackay, Eng Wui Tan and David S. Warren

Our response to the reviewers is mainly focused around comments by reviewer #1.

This is an interesting paper, but it suffers from a very serious lack of organization. In its present form it is merely a report of an interesting program, but if properly organized it could be helpful to other universities that might be interested in organizing their outreach efforts in a similar manner. I think that the article needs a major revision, and that it should be organized to include the following sections:

1. *Introduction*

1.1. *The need for public outreach*

1.2. *Barriers to faculty participation*

1.3. *Emerging trends in science communication*

1.4. *Brief description of the unique aspects of this program*

2. *Methods*

Detailed description of a typical year of this program, including personnel responsibilities, costs, organizational diagrams, etc.

3. *Results*

Outcomes from a year of this program, including number of school visits, number of children served, and number of students, faculty, and staff involved in this effort.

4. *Discussion*

5. *Future directions*

We have re-written the article in line with these recommendations. Given the number of references, especially those supporting our program, combined with the word count we excluded writing "Emerging trends in science communication". We feel that this section in itself would provide enough material for an article in its own right, readers who are interested in this area will find that the references provided will point them in that direction.

In the results section we have expanded on the suggestions by the reviewer and included research evidence from our program showing the impact it has on our students, since we feel the new format lends itself to this approach and this provides a "*raison d'être*" programs of this type. (This also answers the comments by reviewer #2 about primary data).

We found it difficult to know how to structure the manuscript originally, as we have mentioned this is intended for chemists/scientists who are interested in this field. Knowing what would be of interest to them was difficult to predict and we are very grateful to this reviewer for the direction in which they pointed us. We feel it has allowed us to write the article we had originally intended to write and their feedback has allowed us to produce a manuscript that meets our original aims.

In response to reviewer #2

In using the comments from reviewer #1 to re-write the manuscript we feel there are less comments required for reviewer #2, since some of their concerns are addressed by this approach.

We do feel that this reviewer missed the point of the manuscript a little with their focus on the field of science communication. We have worked with science communicators and educational researchers over the lifetime of our program. It was realized, very early on, that if we focus on the impact and learning for our own students then the delivery and impact in the schools would naturally follow on. We hope it can be seen from this new manuscript that we are grounded in educational outcomes (including the development of pedagogy), this is the reason the phrase 'scientist communicators' is in the title and not 'science communicators'. This program runs as a response to new skills sets that are being required of researchers. This being the reason we had originally included literature from social media that reflected the attitudes and confusion in academia around this topic. However, with the expanded format we have included more peer reviewed literature as suggested by this reviewer.

We are grateful to this reviewer for drawing our attention to the work by Louise Archer and the APSIRES study in the UK as well as suggestions for other sources and have included two references from NZ as they suggest that outline what is happening in this country. Although we agree that the program is similar to some that run in the UK (I have visited both at Bristol and Southampton Universities in the past) there are few examples in the literature that give details about how these programs work, or their structure. This article was always intended to generate a discussion of such programs and help identify how they could be set up and run. It was never meant to be a comprehensive study of the science communication literature, however given the increased size of the article and recommendations by reviewer#2 we have included such literature as we feel provides background and evidence for what we do at Otago. We still feel that our definition of outreach, although not exactly in line with some of the literature (e.g. Rowe and Frewer) describes **our** focus and is simple enough that it can be used as the basis for an outreach program run through a chemistry/science department, especially given the variation in the literature. We would anticipate that every outreach program currently running around the world has slightly different criteria and would generate their own definitions of outreach.

.

Yours,

D Waite

REVIEWERS' COMMENTS:

Reviewer #1 (Remarks to the Author):

I am satisfied with the changes to the manuscript. It reads very well, and I think it will be a useful addition to the literature of the field.